# On the possibility of oscillating in the Ebola virus dynamics and investigating the effect of the lifetime of T lymphocytes

**Mehrdad Ghaemi[1]\*, Mina Shojafar[1], Zahra Zabihinpour[2], Yazdan Asgari [3]\***

**1** Department of Chemistry, Kharazmi University, Tehran, Iran, **2** School of Biological Sciences, Institute for Research in Fundamental Sciences (IPM), Tehran, Iran, **3** Department of Medical Biotechnology, School of Advanced Technologies in Medicine, Tehran University of Medical Sciences, Tehran, Iran

\* ghaemi@khu.ac.ir (MG); yasgari@tums.ac.ir (YA)

## Abstract

Ebola virus (EBOV) targets immune cells and tries to inactivate dendritic cells and interferon molecules to continue its replication process. Since EBOV detailed mechanism has not been identified so far, it would be useful to understand the growth and spread of EBOV dynamics based on mathematical methods and simulation approaches. Computational approaches such as Cellular Automata (CA) have the advantage of simplicity over solving complicated differential equations. The spread of Ebola virus in lymph nodes is studied using a simplified Cellular Automata model with only four parameters. In addition to considering healthy and infected cells, this paper also considers T lymphocytes as well as cell movement ability during the simulation in order to investigate different scenarios in the dynamics of an EBOV system. It is shown that the value of the probability of death of T cells affects the number of infected cells significantly in the steady-state. For a special case of parameters set, the system shows oscillating dynamics. The results were in good agreement with an ordinary differential equation-based model which indicated CA method in combination with experimental discoveries could help biologists find out more about the EBOV mechanism and hopefully to control the disease.

## Introduction

Viruses propagate through entering a cell and hijacking the genetic replication machinery in order to make more copies. Once the cell has continued its usual function, the newly manufactured viruses exit to infect other cells. Normally, the human immune system could be able to identify infected cells and destroy them, but most viruses use different strategies to protect their host cells from the immune system. Ebola virus (EBOV) uses a similar method for infection. It targets multiple types of immune cells including dendritic cells (which normally spread alarm infections signals to T lymphocytes) and inactivates them. It also inhibits interferon molecules to continue its replication process [1]. EBOV detailed mechanism has not been identified so far. However, it was uncovered that EBOV encodes for two glycoproteins, one of which disturbs cell attachment [2]. It was previously found that EBOV affects critical pathways related to lymphocytes, such as a specific signalling pathway following binding to a receptor called TLR4 [1]. Experiments suggest that Ebola-infected cells demonstrate fewer proteins on the cell surface, which are critical for an immune recognition process. EBOV belongs to the Filoviruses family first observed in Sudan and Zaire in 1976 and spread to other countries like

**Data Availability Statement:** All relevant data and codes are within the paper and its Supporting Information files.

**Funding:** The authors received no specific funding for this work.

**Competing interests:** The authors have declared that no competing interests exist.

Uganda and Congo [3]. Ebola viral infections are extremely dangerous, with a death rate near 90% for some specific variants. EBOV causes severe bleeding inside and outside the body. So, it has been initially known as the hemorrhagic fever. Although there are some EBOV vaccines and cures that have been established in different stages of clinical trials, none of them has received final approval yet. Therefore, it seems that understanding how EBOV inactivates the immune systems of a host would be a very essential step in developing targeted cures [4].

To understand the growth and spread of viruses and the damage that comes to various tissues, numerous mathematical methods and simulation approaches have been used [5–8]. Most simulation methods of diseases are based on ordinary differential equations (ODEs). In a recent study, Tulu et al. developed a set of ODEs to investigate the dynamics of EBOV spreading [6]. Using mathematical modelling, they demonstrate it is possible to control EBOV spreading by an increment in vaccination. In another work, Wester et al. considered T lymphocytes as targeting molecules in their mathematical model [7]. In addition to the complexity of ODEs-based models, there are still some limitations such as covering local characteristics for transmission of diseases, and ignoring population density variables and population dynamics. Therefore, it is useful to consider complementary simulation methods like Cellular Automata (CA) in order to Fig out more information about virus infection processes. Cellular Automata is a discrete model that could simulate various dynamical systems based on simple rules and local interactions. This model requires an initial configuration and a set of rules that develop the system over time. The central features of CA are a discrete space (one, two, or three-dimensional network), discrete modes (different states for every node), local deterministic or probabilistic rules for every node which only depend on the configuration of adjacent sites, discrete dynamics (renew processes), and homogeneity (similar rules for all sites at every time step) [9]. There are lots of studies using CA approaches on different physical systems such as Ising and Potts models, percolation study, and pattern formation, as well as various biological studies including cancer development, tumor growth, the effect of an anti-cancer drug, angiogenesis, and environmental factors [10–18]. There were also some attempts to simulate Ebola infection using CA method [19, 20]. However, in all CA simulation studies and most other EBOV modelling approaches, healthy, infected, and dead cells have been considered, while T lymphocytes (T cells) are the initial targets for EBOV. Since T cells move through the organs, it could be important if a simulation shows what happens to them during an EBOV infection.

In addition to experimental studies, various computational methods (including neural networks, Monte Carlo, molecular dynamics, and ODEs) have been performed to simulate the dynamics of an EBOV system [6, 7, 21, 22]. However, computational approaches such as CA have the advantage of simplicity over solving complicated differential equations.

This paper uses a CA method in a square lattice for the simulation of EBOV dynamics. Moreover, it also considers T cells as well as cell movement ability during the simulation and investigates different scenarios in the dynamics of an EBOV system.

## Material and methods

We have considered the mathematical model with an ODE-based approach used by Wester et al. [7] in order to obtain appropriate transition rules for our CA model.

$$\frac{dX}{dt} = \lambda - \mu X(t) - \beta V(t)X(t)$$

$$\frac{dI}{dt} = \beta V(t)X(t) - \rho I(t)T(t) - \alpha I(t)$$

$$\frac{dV}{dt} = cI(t) - \gamma V(t) \tag{1}$$

$$\frac{dT}{dt} = \rho I(t)T(t) - \delta T(t)$$

where $X$, $I$, $T$, and $V$ are healthy, infected, T cells, and virus, respectively. $\lambda$ is the growth rate of healthy cells, $\mu$ is the death rate of healthy cells, $\beta$ is healthy cells-virus interaction rate which is the only way to produce infected cells because the immune system does not naturally create infected cells, $\rho$ is infected-T cells interaction rate which shows the activation rate of CTL, and production of lymphocytes, $\alpha$ is the death rate of infected cells. In fact, the first cells target for the virus is lymphocytes. $c$ is the growth rate of the virus, $\gamma$ is the death rate of the virus, and $\delta$ is the death rate of T cells. As it was mentioned in some experimental studies, the most noticeable responses were seen in CD8 T cells (over 50% on activation and proliferation) [23]. Therefore, when we talk about T cells in our model, in fact we consider CD8 types of T cells.

We used a $120 \times 120$ square network as lymphatic tissue with a periodic boundary condition. Increasing the size to $415 \times 415$ will not affect the results. In CA simulation, we have considered only three types of cells (healthy, infected, and T lymphocyte cells). We did not consider a separate situation for dead cells. The Von Neumann neighborhood is used because of its simplicity. Each site of the network just contained one cell at each time. The initial state of all simulation runs consists of only healthy and T cells where the one infected cell is placed at the centre of the network and lymphocytes T cells are distributed randomly with a probability of 50% in the network. Although all the samples have the same number of healthy and T cells and one infected cell at the centre in their initial state, for each sample a different random seed is used, and the initial location of T cells differs for different samples. It should be noted that due to a very small number of infected cells at initial state (only one infected cell) the number of infected cells might go to zero very fast in some samples. We excluded that kind of samples from measurement.

Transition rules include two steps, the stage in which the state of each cell will change (reaction step), and the stage in which each cell moves to one of its adjacent sites (diffusion step).

## Reaction step

Here are the general transition rules:

- In the case of a healthy cell in the site,

○ if at least one infected cell is in its von Neumann vicinity, it will turn to an infected cell with a probability of infection $P_I$, or remains healthy with $(1-P_I)$ probability in the next time step. $P_I$ simulates the probability of transmission of an infection to a healthy cell, which plays the same role as to parameter $\beta$ in Eq (1).

○ if there is not any infected cell in its von Neumann vicinity, it remains healthy.

- In the case of an infected cell in the site,

○ if at least one T cell is in its von Neumann vicinity, it will die and will be replaced by a T cell with $P_T$ probability, or turn to a healthy cell with $(1-P_T)$ probability in the next time step. So $P_T$ represents the probability of T cell proliferation in response to interaction with an infected cell and $(1-P_T)$ represents the probability of the replacement of a dead target cell with a healthy cell. $P_T$ is similar to parameter $\rho$ in Eq (1).

○ if there is not any T cell in its von Neumann vicinity, it will turn to a healthy cell with $P_H$ probability, or remains infected with $(1-P_H)$ probability in the next time step. $P_H$ simulates the probability of natural death of the infected cell, which is equivalent to parameter $\alpha$ in Eq (1).

- In the case of a T cell in the site, it will die and turn to a healthy cell with $P_D$ probability, or remains unchanged with $(1-P_D)$ probability in the next time step. $P_D$ simulates the probability of death of T lymphocyte cells, which plays the same role as parameter $\delta$ in Eq (1).

**Table 1. The probabilities definition.**

| Probability | Definition |
|:---:|:---|
| $P_I$ | Probability of a healthy cell being infected |
| $P_T$ | Probability of a T cell creation |
| $P_H$ | Probability of an infected cell being dead |
| $P_D$ | Probability of a T cell to be dead |

In this simulation, we have considered four probabilities $P_I$, $P_T$, $P_H$, and $P_D$. The definition of these probabilities is given in Table 1.

## Diffusion step

For entering cell movement ability, we used an algorithm similar to the method introduced by Vanag et al. [24]. First, we divided the network into 3×3 blocks and the numbers of healthy, infected, and T cells in each block were counted. Then, the cells were randomly redistributed in the block whereas there was only one cell in each site. It should be noted that after the distribution step, the total number of each cell type remains unchanged in every block. After each time step, all blocks shifted one site in an ordered way; one column to the right, then one row to the bottom, one column to the left, and one row to the top (Fig 1).

## Results and discussion

For obtaining more accurate results, we run the simulation over 50 different samples with the same initial condition, one infected cell at the centre of lattice and equal population of healthy and T cells, and the average number of cells is computed over 50 different samples.

First, we set $P_I = 0.3$, $P_T = 0.4$, $P_H = 0.4$, and $P_D = 0.12$ during the simulation. Fig 2 shows the snapshots of this situation in different time steps. Due to the parameter values, the probability of infected cells being dead and replaced with healthy cells ($P_H$) are more than their creation ($P_I$), nevertheless the number of infected cells in the steady state goes to non-zero value (Fig 3). Initial decreasing of the number of T cells is the result of their natural death and small number of infected cells at the start of simulation. Therefore, the number of T cells increases by increasing the number of infected cells.

Fig 4 shows the dynamics of the system for the case of $P_I = 0.4$, $P_T = 0.6$, and $P_H = 0.4$ while changing $P_D$. In this situation, by increasing the value of $P_I$ contamination ability of an infection increased and total number of infected cells reached to non-zero values in all cases. The total number of T cell in all cases shows a rapid decrease at first time steps because of the large values of $P_D$ in all cases. According to transition rules, creation of T cells depends on the existence of infected cells. Therefore, by increasing the number of infected cells the number of T cells increases. As it is also seen, by increasing the value of the probability of death of T cells ($P_D$), the number of T cells and infected cells in the steady-state decreases and increases respectively.

We also showed a total number of different cell types as a vector graph in which a vector direction shows temporal evolution while vector magnitude demonstrates variation amount. As it is shown in Fig 5, for the case of $P_I = 0.4$, $P_T = 0.6$, $P_H = 0.4$, and $P_D = 0.24$, the number of infected cells remains constant initially with increasing and decreasing the number of healthy and T cells respectively, but after a few time step it increases and reach to a maximum then decreases and goes to the steady-state. As it is seen in Fig 5B the number of T cells decreases initially, but after a few time steps it increases and goes to the steady-state.

To investigate the effect of the value of $P_H$ (the probability of death of infected cells) on the dynamics of the system, three different cases are compared in Fig 6. In all cases, the number of

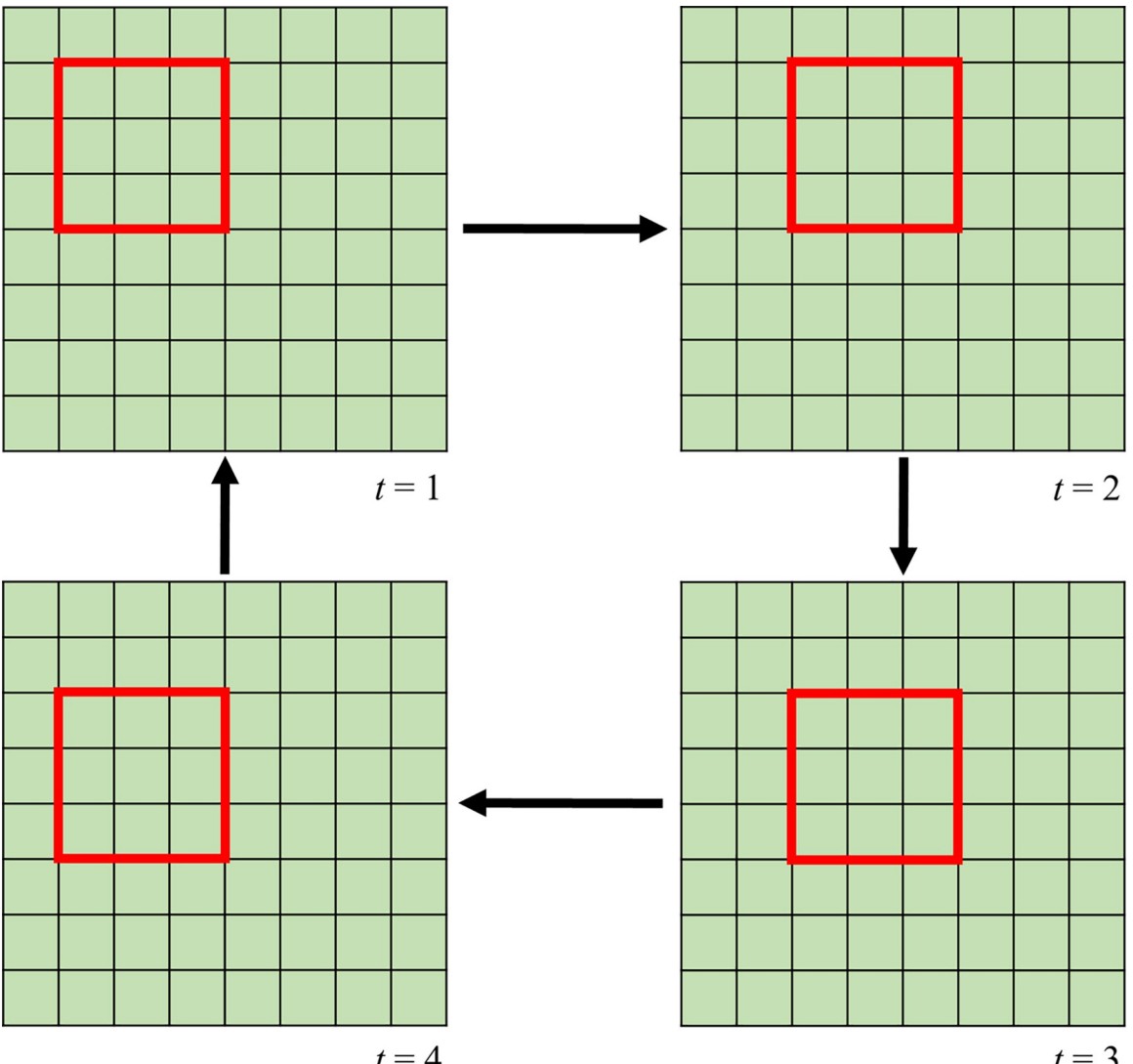

**Fig 1. Schematic view of a block movement in this study in order to consider cell movement ability (see text for more explanation).**

infected cells increased to reach into a steady state. The number of healthy cells shows a rapid growth at the beginning of the automata due to a rapid decrease of the number of T cells because of the large value of $P_D$. After 25 time steps, each infected cell was surrounded approximately by four healthy cells which is the cause of a large number of the healthy cells. Although the value of $P_T$ is greater than the value of $P_I$, the number of T cells is low and a chance of a T cell to come close to infected cells is small. So, the number of T cells increases slowly. By increasing value of the probability of the natural death of the infected cells ($P_H$), the chance of a T cell to join an infected cell before its death decreases. As shown in Fig 6C, in the case of $P_H = 0.9$, the number of the T cells turned to a near zero value after initial decrement and then an increment very smoothly to reach into a steady state after about 300 time steps.

The results in Fig 6 show for this set of parameters, increasing the value of $P_H$ does not affect significantly the number of infected cells in the steady-state. Note in Fig 6C the value of $P_H$ is very larger than the value of $P_I$ nevertheless the number of infected cells reaches a steady-state.

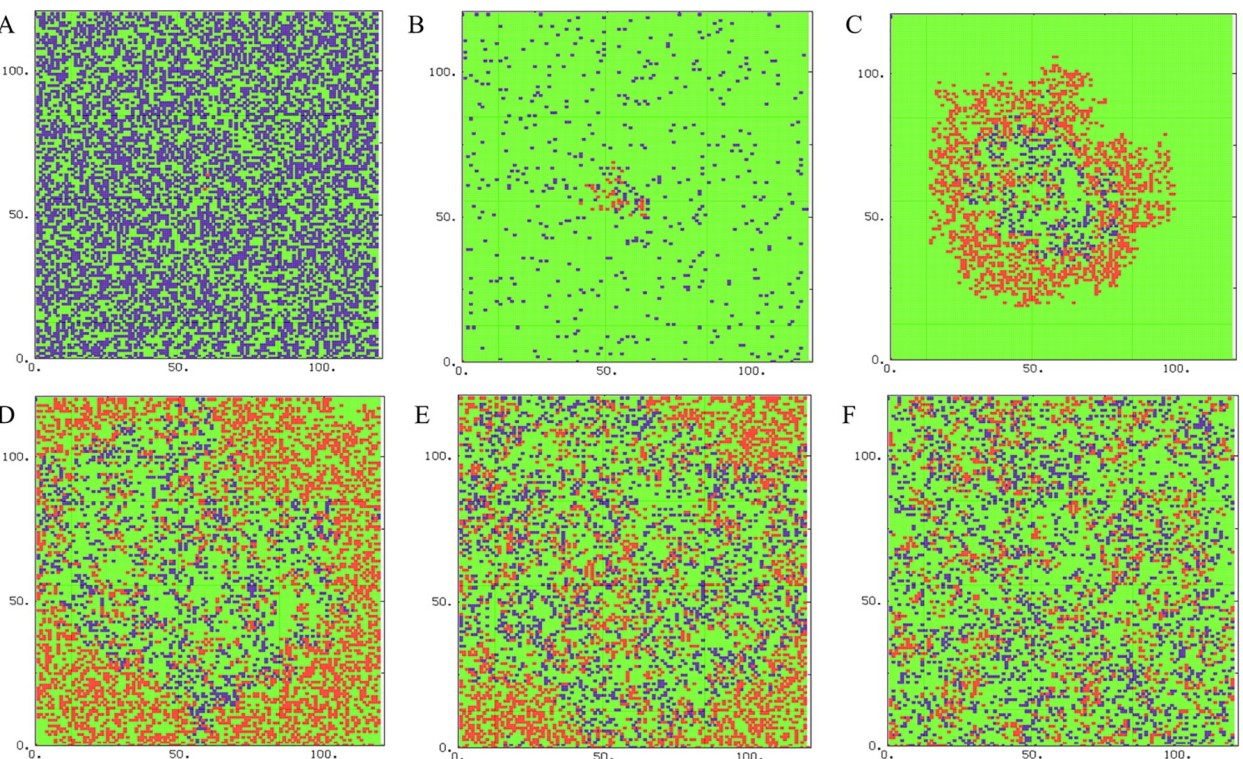

**Fig 2.** The snapshots of CA simulation of one sample for the set of parameters ($P_I$ = 0.3, $P_T$ = 0.4, $P_H$ = 0.4, $P_D$ = 0.12) in different time steps; $t$ = 1 (A), $t$ = 20 (B), $t$ = 60 (C), $t$ = 100 (D), $t$ = 120 (E), and $t$ = 200 (F). Healthy, infected, and T cells are shown in green, red, and blue, respectively.

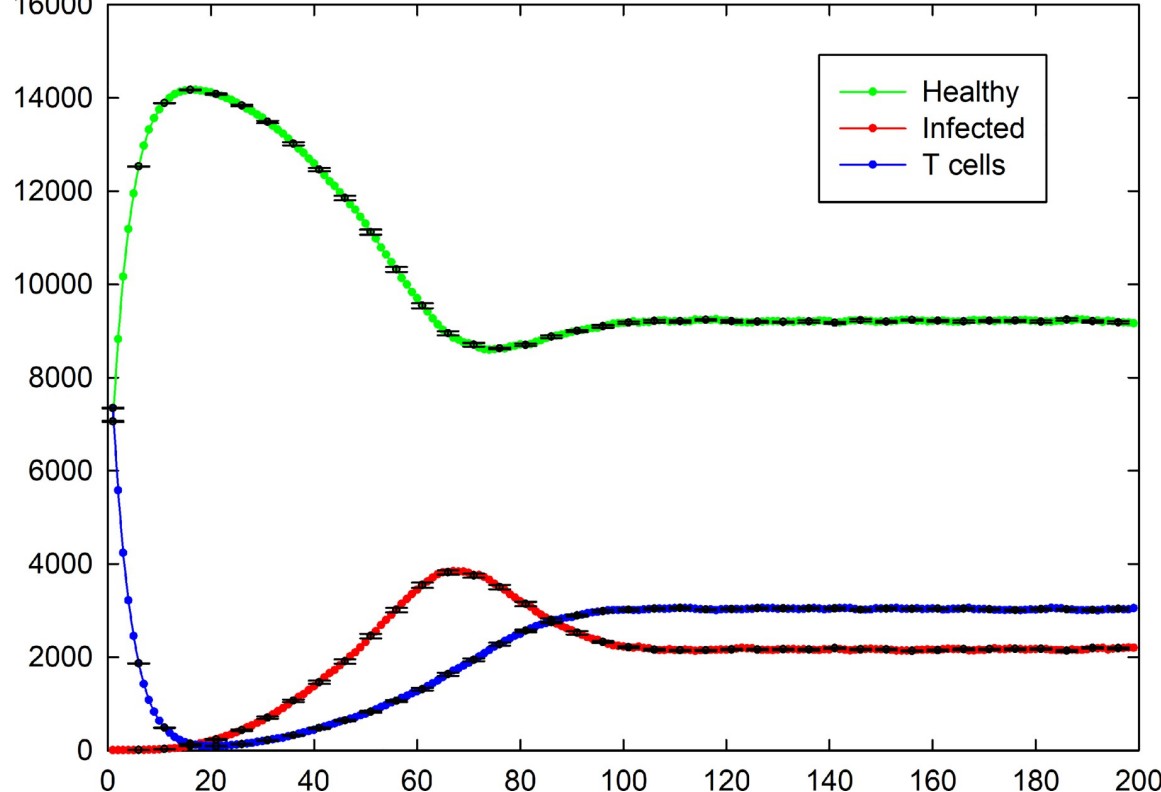

**Fig 3. Total number of healthy, infected, and T cells versus time for the case of $P_I$ = 0.3, $P_T$ = 0.4, $P_H$ = 0.4, $P_D$ = 0.12.** Healthy, infected, and T cells are shown in green, red, and blue, respectively. The standard errors are shown by error bars.

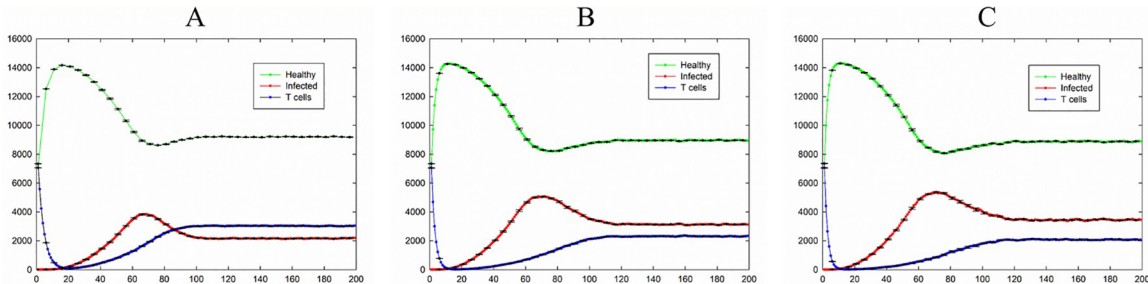

**Fig 4.** Total number of healthy, infected, and T cells versus time for the case of $P_I$ = 0.4, $P_T$ = 0.6, and $P_H$ = 0.4; A) $P_D$ = 0.24. B) $P_D$ = 0.36. C) $P_D$ = 0.4. Healthy, infected, and T cells are shown in green, red, and blue, respectively. The standard errors are shown by error bars.

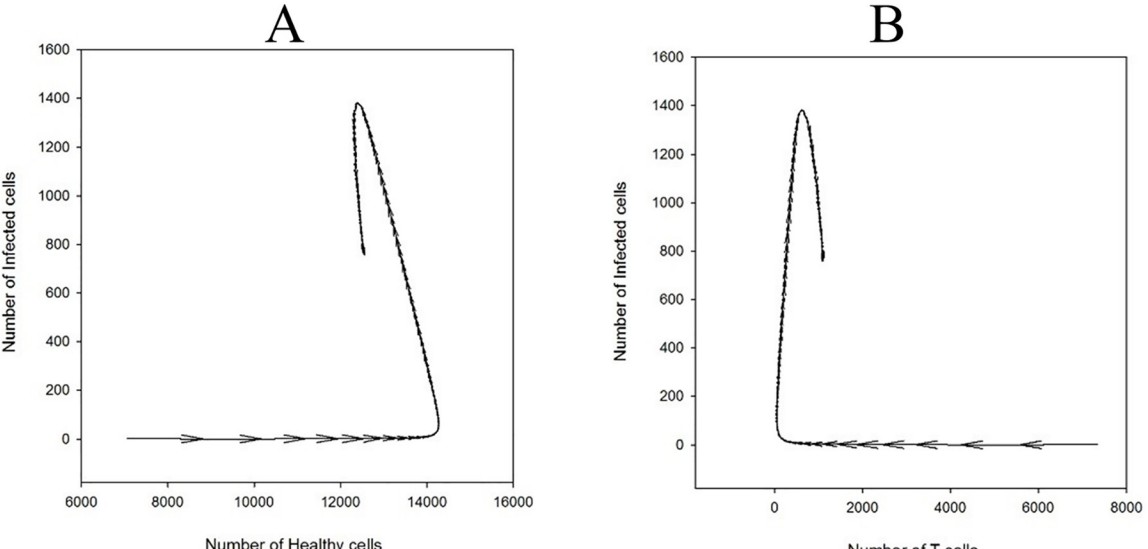

**Fig 5. Total number of different cell types as a vector graph for the case of $P_I$ = 0.4, $P_T$ = 0.6, $P_H$ = 0.4, and $P_D$ = 0.24.** Vector direction shows temporal evolution while vector magnitude demonstrates variation amount: A) The number of infected versus the number of healthy cells, B) The number of infected versus the number of T cells.

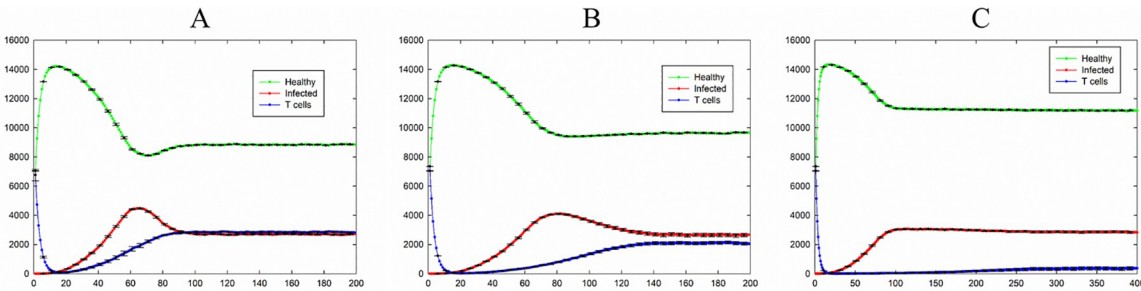

**Fig 6.** Total number of healthy, infected, and T cells versus time for the case of $P_I$ = 0.4, $P_T$ = 0.6, and $P_D$ = 0.3; A) $P_H$ = 0.3. B) $P_H$ = 0.6. C) $P_H$ = 0.9. Healthy, infected, and T cells are shown in green, red, and blue, respectively. The standard errors are shown by error bars.

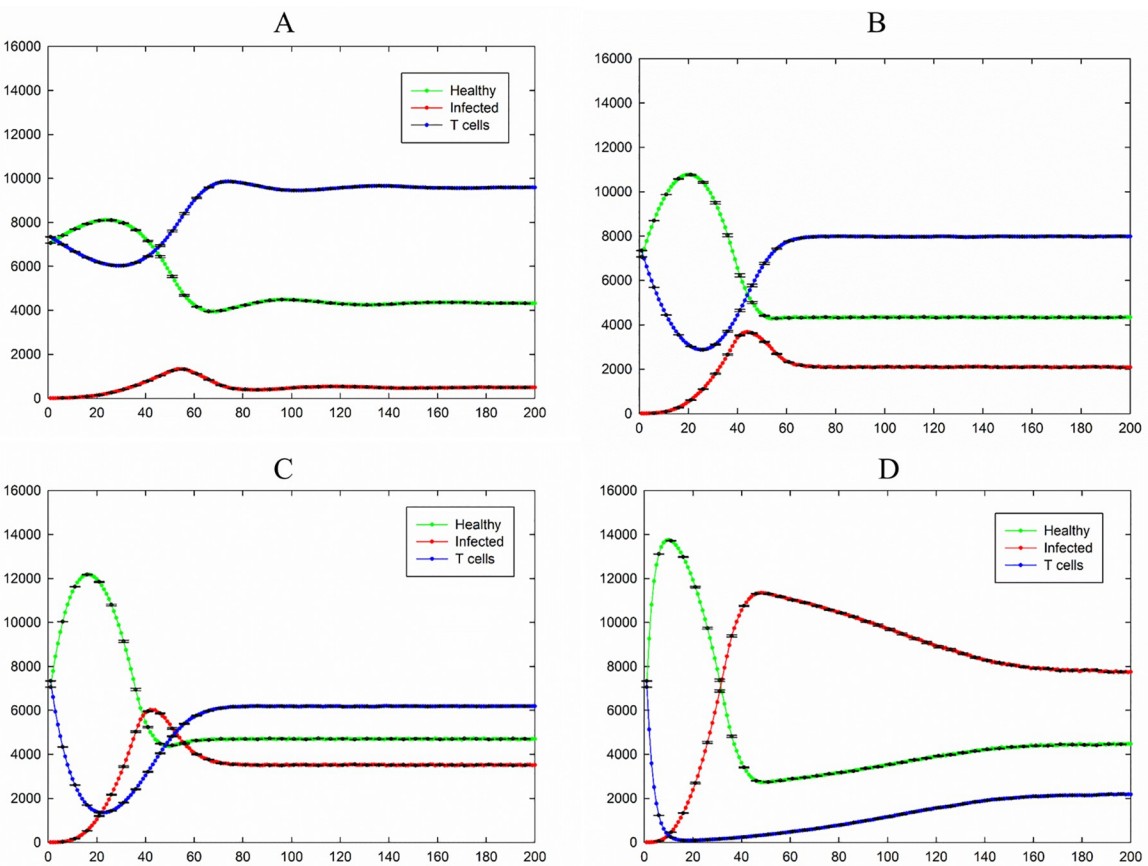

**Fig 7. Total number of healthy, infected, and T cells versus time for the case of $P_I = 1.0$, $P_T = 0.2$, and $P_H = 0.2$.** A) $P_D = 0.01$, B) $P_D = 0.05$, C) $P_D = 0.1$, D) $P_D = 0.3$. Healthy, infected, and T cells are shown in green, red, and blue, respectively. The standard errors are shown by error bars.

To study whether a large number of T cells is responsible for killing infected cells, we set $P_I = 1$ which means the contamination ability of infected cells is at its maximum value. Fig 7 shows simulation results for the case of $P_I = 1$, $P_T = 0.2$, and $P_H = 0.2$ and different values of $P_D$.

For the long time life of T cells (Fig 7A) the number of infected cells after a maximum reach to the steady-state which is small compared with the total number of cells and the number of T cells in the steady-state is larger than the number of healthy cells. As it is shown in Fig 7, by decreasing the time life of T cells (increasing the value of $P_D$) the number of infected and T cells at the steady-state increases and decreases respectively and the number of healthy cells at the steady-state remains approximately unchanged.

For the case of $P_I = 1$, $P_T = 0.4$, $P_H = 0.4$, and $P_D = 0.01$, the results are shown in Fig 8 and all sample included in the measurement. In this special case, the system shows oscillating dynamics. Data analysis shows a large value for standard deviation (for example in an order of 3000 for the number of healthy cells after 500 time steps). Such a large value for standard deviation indicates existence of constant error.

Total number of different cell types versus time for five different samples are shown in Fig 9. As it is seen, not all of individuals oscillate at the same frequency and amplitude. In addition, one of the samples does not show oscillating dynamics and for this sample, the number of infected cells goes to zero very fast due to a small number of this cells at initial state (namely

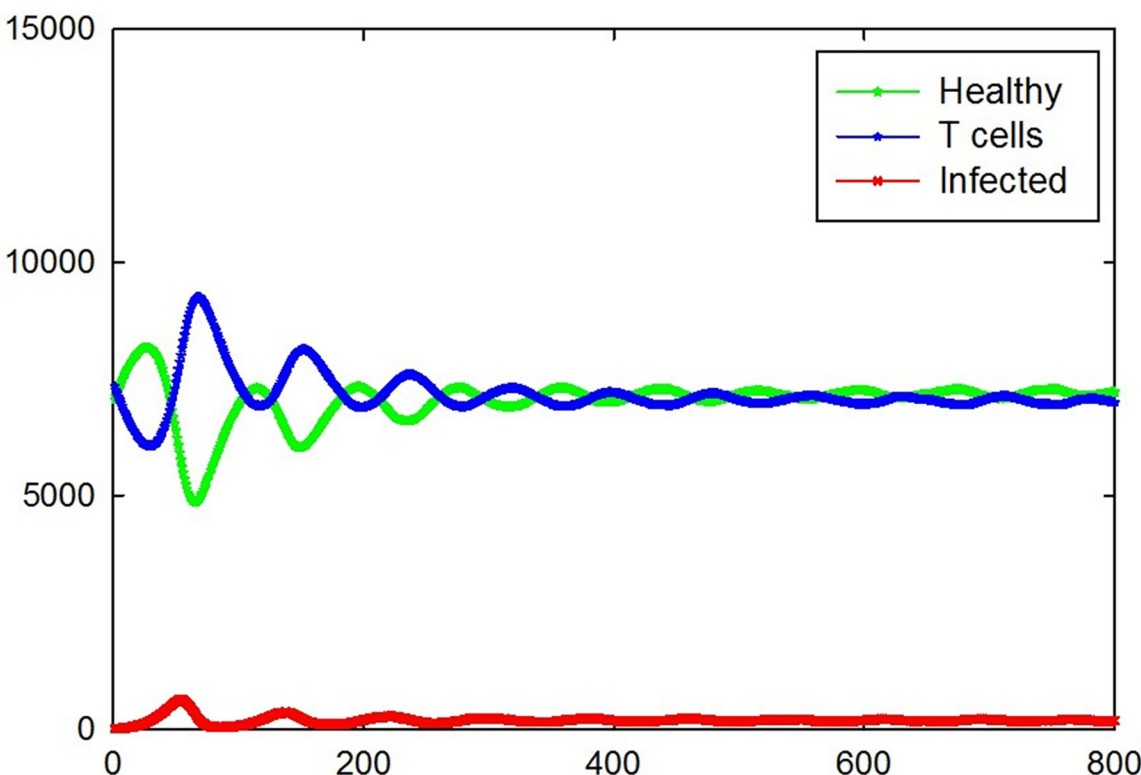

**Fig 8. Total number of healthy, infected, and T cells versus time for the case of $P_I = 1$, $P_T = 0.4$, $P_H = 0.4$, and $P_D = 0.01$.**

only one infected cell) and the number of T cells decreases slowly to a zero value due to the natural death. We removed eleven samples of that kind of situations from the measurement. Therefore, total number of different cell types versus time which were calculated by performing an average over 39 remaining samples, are shown in Fig 10 and their standard errors are shown by error bars. By passing time the domain and frequency of oscillation will decrease and increase respectively and finally reaches a constant value. Similar to the case of Fig 7A, the number of infected cells at the steady-state in this case is very low.

The snapshots of one of the samples which are taken at every 20 time steps are shown in Fig 11. The simulation started by inserting one infected cell in the centre of the lattice. In this case, we set $P_I = 1$ which means the contamination ability of the infected cells is at its maximum value. The ability of movements of the cells makes the chance to the infected cells to come close to the healthy cells and convert them to the infected cells. In contrast, the number of T cells will increase in the centre of the lattice because of the value of the probability of creation of a T cell is much more than the value of the probability of their death. These two mechanisms make the possibility of formation of moving circular waves with infected cells at their boundary and large numbers of T cells at the inner layers. By passing time, the size and the number of these waves decrease and increase respectively.

We also showed a total number of different cell types for this case as a 2-dimentional and 3-dimentional vector graphs in Fig 12A–12C. The oscillation in the number of infected cells is clearer in this figure.

The effect of an increasing value of $P_I$ is shown in Figs 3, 4, 6 and 7D. For $P_I$ larger than 0.3, the number of infected cells in the steady state reached to non-zero values and increased. Comparing Fig 7D with Figs 4 and 6 demonstrates that the number of healthy cells in this case at

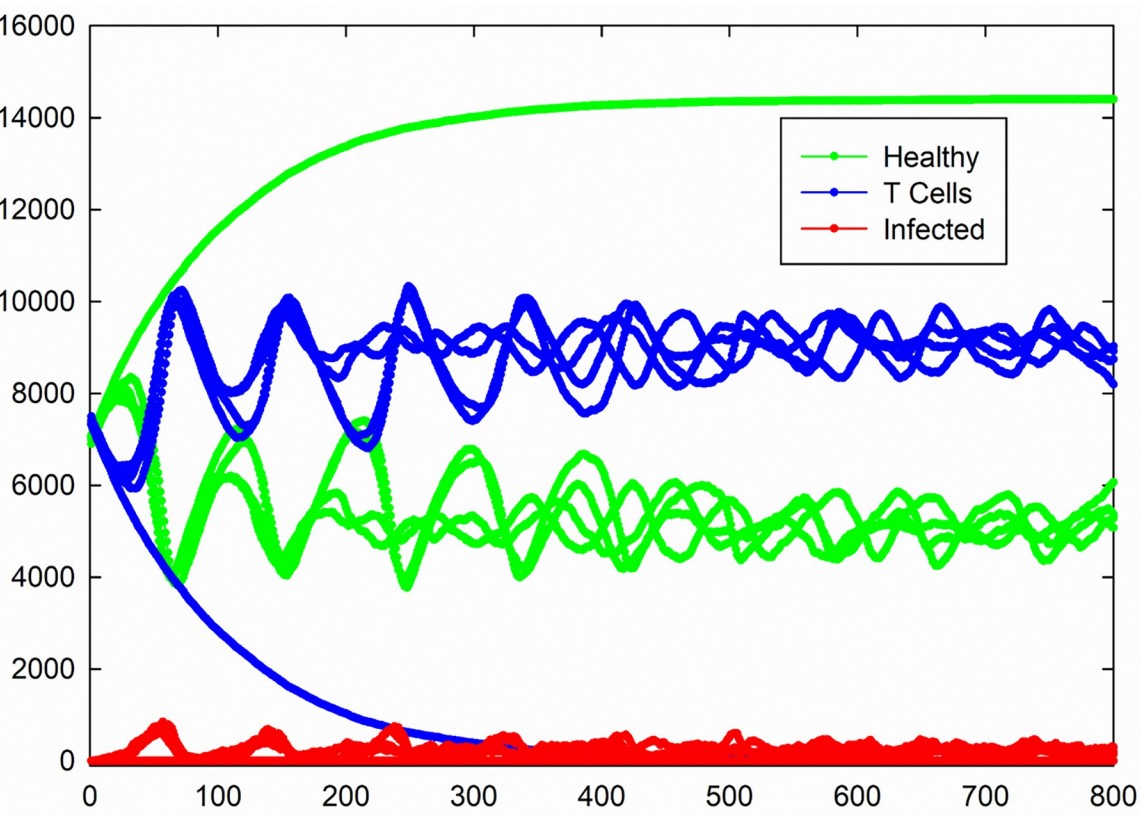

**Fig 9. Total number of healthy, infected, and T cells versus time for 5 different samples for the case of $P_I = 1$, $P_T = 0.4$, $P_H = 0.4$, and $P_D = 0.01$.**

the steady state is much smaller than the previous cases and the number of infected cells increased.

Comparing Fig 7A with Fig 8 shows that increasing the value of $P_T$ has not a significant effect on the number of infected and healthy cells in the steady state. The results show two important factors which are able to decrease the number of infected cells significantly; the values of $P_I$ and $P_D$. As shown in Figs 7A and 8, for the case of $P_I = 1$ (maximum value of contamination ability of infected cells) but the very small value of $P_D$ (long lifetime of T cells), the number of infected cells in the steady-state goes to a very small value. By decreasing the lifetime of T cells, the number of infected cells in the steady state increases (Figs 4 and 7). Increasing the value of the probability of death of infected cells ($P_H$) has not a significantly effect on the number of infected cells in the steady state. It should be noted that in Fig 6C the value of $P_H$ is very larger than the value of $P_I$ nevertheless the number of infected cells does not go to zero in the steady state.

In the above simulations we let each cell move to adjacent sites in each time step. The mechanism of this procedure is defined previously in section 2. Fig 13 shows the dynamics of the system for the case of $P_I = 1$, $P_T = 0.4$, $P_H = 0.4$, and $P_D = 0.01$ without applying diffusion steps in transition rules. In another word, for this situation, each cell has a fixed position and cannot move. For this situation only two samples showed oscillating dynamics and the remained 48 samples passed one cycle of oscillation and some of them passed second cycle of oscillation and presence of that kind of samples made larger standard errors in the number of T cells and healthy cells in the times between 200 to 500 time steps. The number of infected cells at the

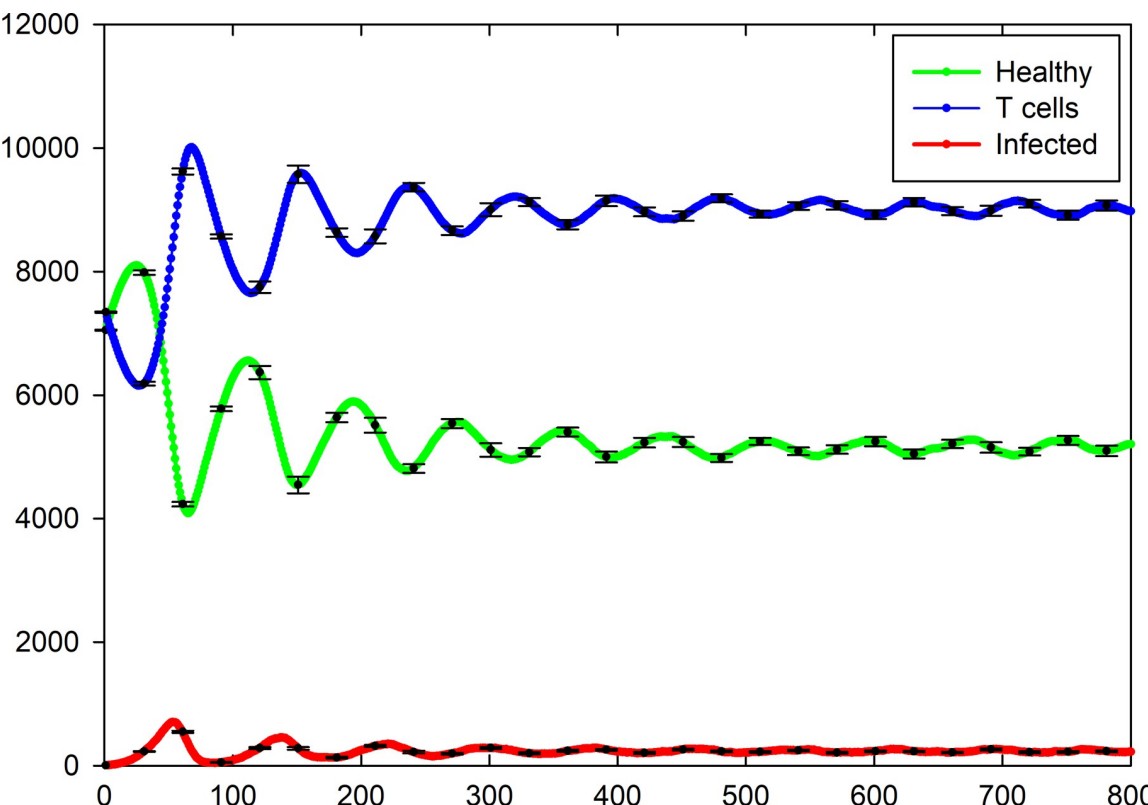

**Fig 10. Total number of healthy, infected, and T cells versus time which are calculated by averaging over 39 samples with oscillating dynamics, for the case of $P_I = 1$, $P_T = 0.4$, $P_H = 0.4$, and $P_D = 0.01$.** The standard errors are shown by error bars.

steady state reaches to a small value in this case but by passing time, the number of T cells and healthy cells changes completely different compared to Fig 10. This result indicates a movement of cells has important contribution in creating oscillating dynamics.

## Conclusion

It is critical to know how the immune system responds to an infection caused by Ebola virus since EBOV survival depends on host cell abilities. In addition to experimental studies, various computational methods (including neural networks, Monte Carlo, molecular dynamics, and ODEs) have been performed to simulate the dynamics of an EBOV system [6, 7, 21, 22]. However, computational approaches such as CA have the advantage of simplicity instead of solving complicated differential equations. Here, we have studied the spread of EBOV in lymph nodes including healthy, infected, and T cells using a simplified CA model with just four probability values. The impairment ability of infected cells was considered with $P_I$ probability, while the probability of a T cell creation was $P_T$. Death probabilities of infected and T cells were $P_H$ and $P_D$ probabilities, respectively. We also compared our results with the results of an ODE model proposed by Wester et al. [7]. In general, they used combination of some parameters for generating their data and because of that, an exact and one by one comparisons of their results with our CA results are not possible. However, some similarity could be found. For example, in Fig 8 of their work, an oscillation dynamics behaviour could be also seen which is in concordance with an oscillation behaviour in the CA model.

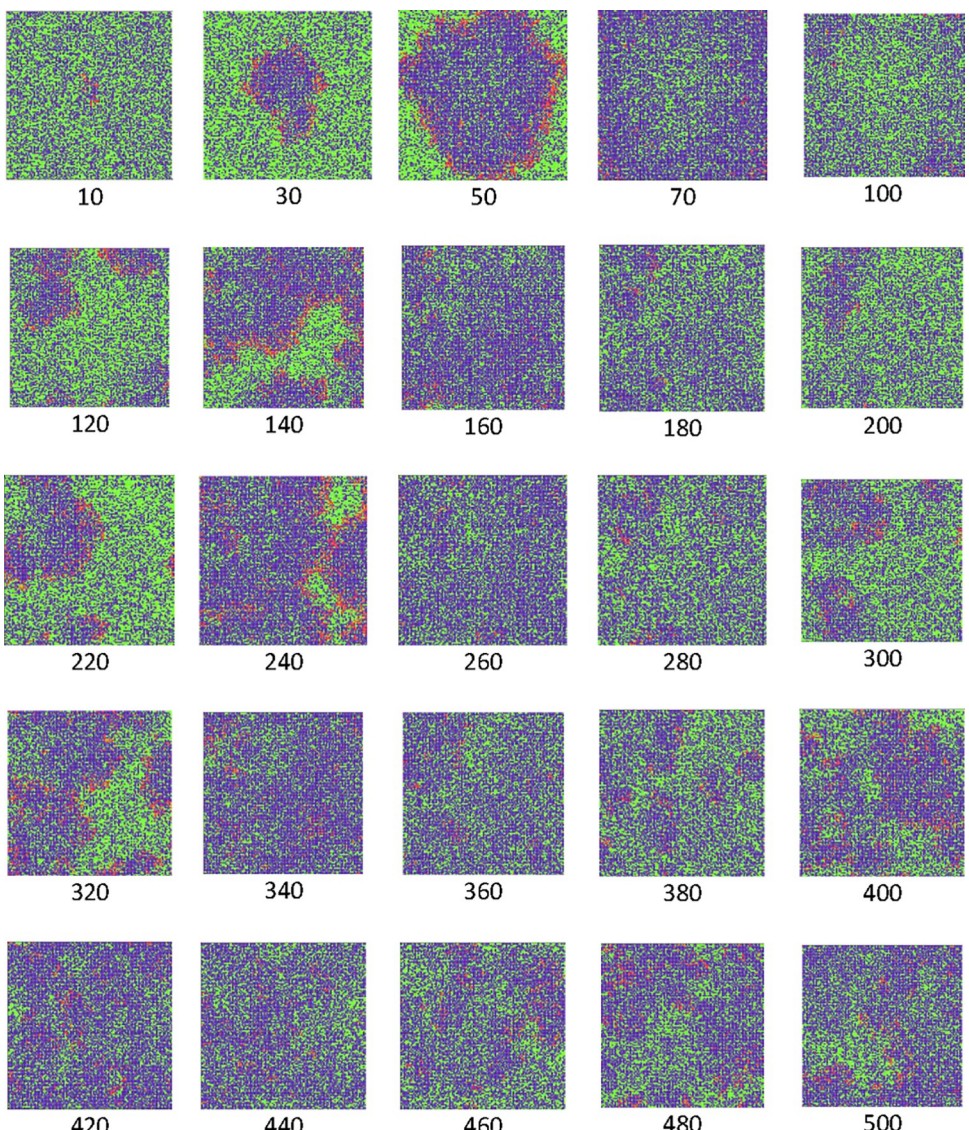

**Fig 11. The snapshots at different time steps for the case of $P_I = 1$, $P_T = 0.4$, $P_H = 0.4$, and $P_D = 0.01$.** Healthy, infected, and T cells are shown in green, red, and blue, respectively.

As far as we know, there is no approved vaccine available for clinical use of Ebola and scientists try to find out more about its mechanism. For instance, they recently showed for the first time that despite the incapability of Ebola to infect lymphocytes, it directly binds to them, involves the TLR4 pathway, and causes cell death [25]. The results demonstrated that adding a chemical that blocks TLR4 activation could protect the lymphocytes in the presence of Ebola. This could lead to design a drug that blocks TLR4 and might be used to treat patients with Ebola. Therefore, such experimental discoveries in combination with simulation methods could lead to a better understanding of EBOV dynamics of operation, its replication, and hopefully to control the disease. Our study revealed using CA methods could be useful in exploring the dynamics of an EBOV system even in the case of considering details whereas avoiding the complexity of ODEs. It is also possible to add new parameters and variables in a

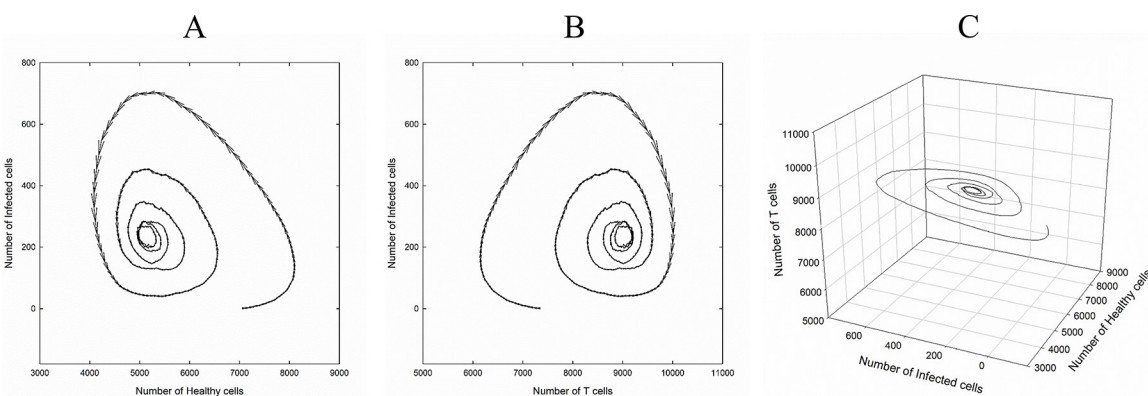

**Fig 12. Total number of different cell types as a vector graph for the case of $P_I = 1$, $P_T = 0.4$, $P_H = 0.4$, and $P_D = 0.01$.** Vector direction shows temporal evolution while vector magnitude demonstrates variation amount: A) The number of infected versus the number of healthy cells, B) The number of infected versus the number of T cells. C) a 3-dimentional graph of the number of infected cells versus the number of healthy and T cells.

CA approach very easily. We believe, in the case of EBOV, CA method could help biologists to find out more about the mechanism since there are still much vaguenesses about the EBOV activity in the human body.

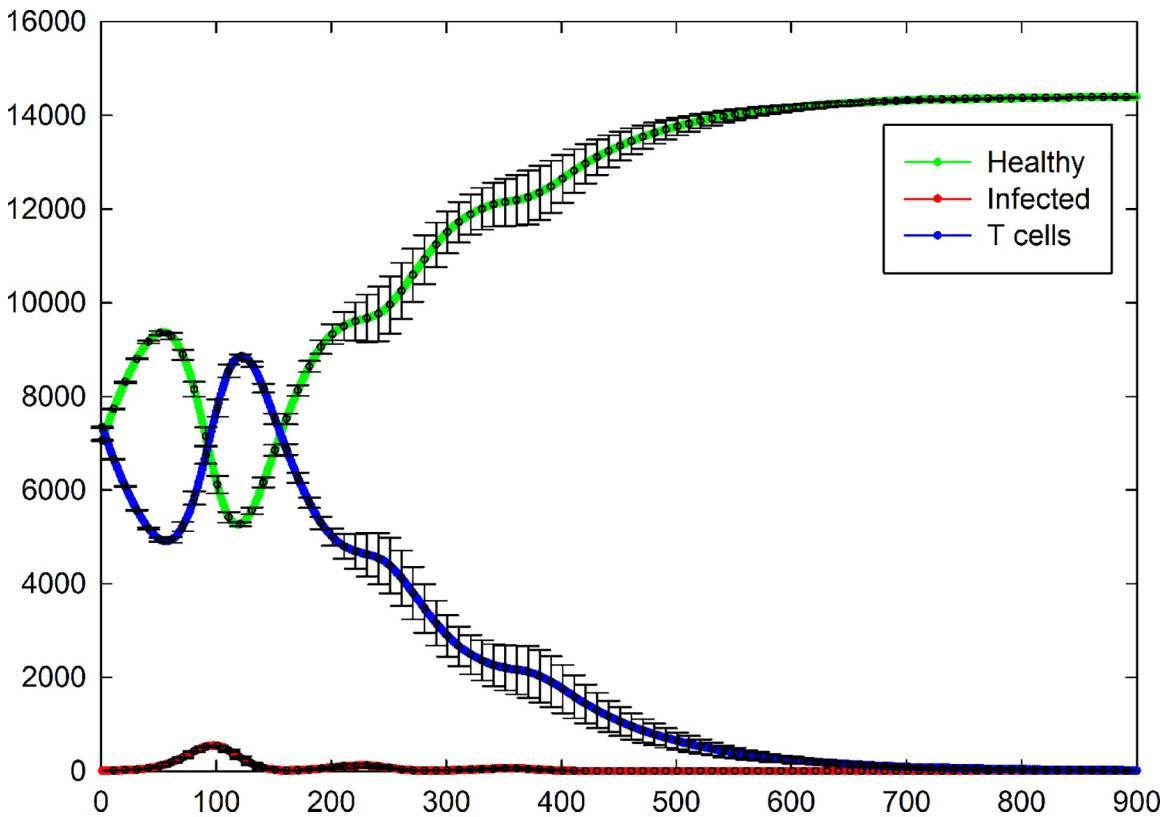

**Fig 13. Total number of healthy, infected, and T cells versus time for the case of $P_I = 1$, $P_T = 0.4$, $P_H = 0.4$, and $P_D = 0.01$ without applying diffusion step in transition rules.** Healthy, infected, and T cells are shown in green, red, and blue, respectively. The standard errors are shown by error bars.

## Supporting information

**S1 File.**
(ZIP)

## Acknowledgments

The authors acknowledgment the reviewers for their useful comments.

## Author Contributions

**Conceptualization:** Mehrdad Ghaemi.

**Data curation:** Mehrdad Ghaemi.

**Formal analysis:** Mehrdad Ghaemi, Mina Shojafar, Zahra Zabihinpour.

**Investigation:** Mehrdad Ghaemi, Yazdan Asgari.

**Methodology:** Mehrdad Ghaemi.

**Project administration:** Mehrdad Ghaemi.

**Software:** Mehrdad Ghaemi, Mina Shojafar.

**Supervision:** Mehrdad Ghaemi.

**Validation:** Mehrdad Ghaemi, Zahra Zabihinpour.

**Visualization:** Mehrdad Ghaemi, Mina Shojafar, Zahra Zabihinpour.

**Writing – original draft:** Mehrdad Ghaemi, Mina Shojafar, Zahra Zabihinpour, Yazdan Asgari.

**Writing – review & editing:** Mehrdad Ghaemi, Zahra Zabihinpour, Yazdan Asgari.

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
