## [Decision Letter · Decision Letter 0]

9 Nov 2021

PONE-D-21-29749On the Possibility of Oscillating in the Ebola Virus Dynamics and Investigating the Effect of the Lifetime of T lymphocytesPLOS ONE

Dear Dr. Asgari,

Thank you for submitting your manuscript to PLOS ONE. After careful consideration, we feel that it has merit but does not fully meet PLOS ONE’s publication criteria as it currently stands. Therefore, we invite you to submit a revised version of the manuscript that addresses the points raised during the review process. The review below raises some concerns about the model and identifies areas in which results from the cellular automata model should be analyzed in more detail. 

We look forward to receiving your revised manuscript.

Kind regards,

Steven M. Abel, Ph.D.

Academic Editor

PLOS ONE

Journal Requirements:

2. Please note that PLOS ONE has specific guidelines on code sharing for submissions in which author-generated code underpins the findings in the manuscript. In these cases, all author-generated code must be made available without restrictions upon publication of the work. Please review our guidelines at https://journals.plos.org/plosone/s/materials-and-software-sharing#loc-sharing-code and ensure that your code is shared in a way that follows best practice and facilitates reproducibility and reuse

Reviewers' comments:

Reviewer's Responses to Questions

**Comments to the Author**

1. Is the manuscript technically sound, and do the data support the conclusions?

Reviewer #1: Partly

2. Has the statistical analysis been performed appropriately and rigorously? 

Reviewer #1: N/A

3. Have the authors made all data underlying the findings in their manuscript fully available?

Reviewer #1: No

4. Is the manuscript presented in an intelligible fashion and written in standard English?

Reviewer #1: Yes

5. Review Comments to the Author

Reviewer #1: The authors of “On the Possibility of Oscillating in the Ebola Virus Dynamics and Investigating the Effect of the Lifetime of T lymphocytes” describe a cellular automata (CA) model to represent the spread of Ebola virus in lymph nodes.

While this is an important topic and the authors present interesting results, there are some questions and concerns about the model, as well as potentially some missed opportunities to take advantage of what CA models can offer.

1. One strength of CA models is to offer spatial insights, yet there are no spatial results given beyond figure 2. One important question arising from their results is: what does the spatial distribution of infected cells look like in simulations that oscillate?

2. Another missed opportunity lies in comparing results between the ODE and CA models (which the authors state that they will do (p4), and mention again on p16) but do not present any results for). It would be interesting to see if the oscillations arise because of the spatial nature of CA models, or if this is something that is preserved from the ODE model?

3. Another key feature of CA models is their stochasticity. The authors write that they average 50 simulations, which raises a few questions

a. Why not include standard error/standard deviation in their figures? Only showing the average does not represent the full range of possible outcomes and makes it difficult to evaluate the impact of stochasticity in this system.

b. When they report oscillations in their average values (e.g. Fig 8) – does that imply that all of the individual simulations oscillate at the same frequency and amplitude?

4. In the introduction, the authors said that T cells are initial targets. But healthy cells which can be infected and T cells are two different groups in their model. Infected cells with T cells around can change either to healthy cells or T cells. It's unclear what is the relationship (and the difference) between healthy cells and T cells in the model.

5. There are many different types if T cells with different functions. Please specify the type of T cell considered in this model.

6. In the results section a lot of different scenarios are tested, with different parameters and movement settings, which is good. But some problems exist:

a. Influence of PH and PD are specifically tested with different values, but it is unclear why those two are chosen, not PI and PT.

b. PI and PT are also changing from fig. 3 to fig. 8. But the change is not specifically mentioned. What is the influence of these parameters?

c. In fig. 6 C, when the death rate of infected cells is high, instead of infected cells, T cells die out. This is quite counter intuitive and warrants more discussion. Also, it is surprising that infected cells can be maintained at low level in the absence of T cells.

d. There are lots of details given describing what the figures show, but this type of discussion distracts from the main idea that's being delivered. There should be some more conclusive statement summarizing/synthesizing the findings from each scenario.

e. The oscillations are a key feature of their results, but an explanation of the source of oscillations is lacking.

7. The second paragraph in the results section mostly summarizes their results again. Some of this discussion would be better included in the results section as it doesn’t really help the reader connect their findings to the broader field.

6. PLOS authors have the option to publish the peer review history of their article (what does this mean?). If published, this will include your full peer review and any attached files.

Reviewer #1: No

---

## [Author Response · Author response to Decision Letter 0]

6 Jan 2022

We thank the reviewers for their detailed comments on our submitted manuscript. We submitted our point by point response to address Reviewers’ comments as a separate document.

---

## [Decision Letter · Decision Letter 1]

18 Feb 2022

PONE-D-21-29749R1On the Possibility of Oscillating in the Ebola Virus Dynamics and Investigating the Effect of the Lifetime of T lymphocytesPLOS ONE

Dear Dr. asgari,

Thank you for submitting your manuscript to PLOS ONE. After careful consideration, we feel that it has merit but does not fully meet PLOS ONE’s publication criteria as it currently stands. Therefore, we invite you to submit a revised version of the manuscript that addresses the minor concerns raised below.

We look forward to receiving your revised manuscript.

Kind regards,

Steven M. Abel, Ph.D.

Academic Editor

PLOS ONE

Journal Requirements:

Reviewers' comments:

Reviewer's Responses to Questions

**Comments to the Author**

1. If the authors have adequately addressed your comments raised in a previous round of review and you feel that this manuscript is now acceptable for publication, you may indicate that here to bypass the “Comments to the Author” section, enter your conflict of interest statement in the “Confidential to Editor” section, and submit your "Accept" recommendation.

Reviewer #1: (No Response)

2. Is the manuscript technically sound, and do the data support the conclusions?

Reviewer #1: Yes

3. Has the statistical analysis been performed appropriately and rigorously? 

Reviewer #1: Yes

4. Have the authors made all data underlying the findings in their manuscript fully available?

Reviewer #1: Yes

5. Is the manuscript presented in an intelligible fashion and written in standard English?

Reviewer #1: Yes

6. Review Comments to the Author

Reviewer #1: The authors have made significant updates to improve the manuscript. Two minor comments based on their revisions are below.

The addition of figure 9 is very helpful. The authors explain the cases where the infected cells go to zero as being because of starting with only one infected cell. But don’t all simulations start with one infected cell?

It is still not clear to me what the parameter PT represents and what it means biologically. Does it represent T cell proliferation in response to interaction with an infected cell? Does (1-PT) represent regeneration of the dead target cell?

7. PLOS authors have the option to publish the peer review history of their article (what does this mean?). If published, this will include your full peer review and any attached files.

Reviewer #1: No

---

## [Author Response · Author response to Decision Letter 1]

21 Feb 2022

We have responded to the reviewers’ comments point-by-point and uploaded this as a separate file labeled 'Response to Reviewers'.

---

## [Editor Report · Decision Letter 2]

23 Feb 2022

On the Possibility of Oscillating in the Ebola Virus Dynamics and Investigating the Effect of the Lifetime of T lymphocytes

PONE-D-21-29749R2

Dear Dr. asgari,

We’re pleased to inform you that your manuscript has been judged scientifically suitable for publication and will be formally accepted for publication once it meets all outstanding technical requirements.

Kind regards,

Steven M. Abel, Ph.D.

Academic Editor

PLOS ONE
---

## [Editor Report · Acceptance letter]

1 Mar 2022

PONE-D-21-29749R2 

On the Possibility of Oscillating in the Ebola Virus Dynamics and Investigating the Effect of the Lifetime of T lymphocytes 

Dear Dr. asgari:

I'm pleased to inform you that your manuscript has been deemed suitable for publication in PLOS ONE. Congratulations! Your manuscript is now with our production department. 

Kind regards, 

on behalf of

Dr. Steven M. Abel 

Academic Editor

PLOS ONE